# Metabolomic Insight into Implications of Induction Chemotherapy Followed by Concomitant Chemoradiotherapy in Locally Advanced Head and Neck Cancer

**DOI:** 10.3390/ijms25010188

**Published:** 2023-12-22

**Authors:** Łukasz Boguszewicz, Agata Bieleń, Mateusz Ciszek, Agnieszka Skorupa, Jolanta Mrochem-Kwarciak, Krzysztof Składowski, Maria Sokół

**Affiliations:** 1Department of Medical Physics, Maria Sklodowska-Curie National Research Institute of Oncology, Gliwice Branch, 44-102 Gliwice, Poland; mateusz.ciszek@gliwice.nio.gov.pl (M.C.); agnieszka.skorupa@gliwice.nio.gov.pl (A.S.); maria.sokol@gliwice.nio.gov.pl (M.S.); 21st Radiation and Clinical Oncology Department, Maria Sklodowska-Curie National Research Institute of Oncology, Gliwice Branch, 44-102 Gliwice, Poland; agata.bielen@gliwice.nio.gov.pl (A.B.);; 3Analytics and Clinical Biochemistry Department, Maria Sklodowska-Curie National Research Institute of Oncology, Gliwice Branch, 44-102 Gliwice, Poland; jolanta.mrochem-kwarciak@gliwice.nio.gov.pl

**Keywords:** head and neck cancer, NMR spectroscopy, metabolomics, induction chemotherapy, chemoradiotherapy, treatment toxicity, LA-HNSCC

## Abstract

The present study compares two groups of locally advanced patients with head and neck squamous cell carcinoma (LA-HNSCC) undergoing concurrent chemoradiotherapy (cCHRT), specifically those for whom it is a first-line treatment and those who have previously received induction chemotherapy (iCHT). The crucial question is whether iCHT is a serious burden during subsequent treatment for LA-HNSCC and how iCHT affects the tolerance to cCHRT. Of the 107 LA-HNSCC patients, 54 received cisplatin-based iCHT prior to cCHRT. The patients were clinically monitored at weekly intervals from the day before until the completion of the cCHRT. The 843 blood samples were collected and divided into two aliquots: for laboratory blood tests and for nuclear magnetic resonance (NMR) spectroscopy (a Bruker 400 MHz spectrometer). The NMR metabolites and the clinical parameters from the laboratory blood tests were analyzed using orthogonal partial least squares analysis (OPLS) and the Mann–Whitney U test (MWU). After iCHT, the patients begin cCHRT with significantly (MWU *p*-value < 0.05) elevated blood serum lipids, betaine, glycine, phosphocholine, and reticulocyte count, as well as significantly lowered NMR inflammatory markers, serine, hematocrit, neutrophile, monocyte, red blood cells, hemoglobin, and CRP. During cCHRT, a significant increase in albumin and psychological distress was observed, as well as a significant decrease in platelet, N-acetyl-cysteine, tyrosine, and phenylalanine, in patients who received iCHT. Importantly, all clinical symptoms (except the decreased platelets) and most metabolic alterations (except for betaine, serine, tyrosine, glucose, and phosphocholine) resolve until the completion of cCHRT. In conclusion, iCHT results in hematological toxicity, altered lipids, and one-carbon metabolism, as well as downregulated inflammation, as observed at the beginning and during cCHRT. However, these complications are temporary, and most of them resolve at the end of the treatment. This suggests that iCHT prior to cCHRT does not pose a significant burden and should be considered as a safe treatment option for LA-HNSCC.

## 1. Introduction

Head and neck squamous cell carcinomas (HNSCC) develop in organs that play pivotal roles in respiratory, nutritional and social functions. Therefore, the first goal of a locally advanced HNSCC (LA-HNSCC) treatment strategy is to protect these organs. The second goal is the debulking strategy; when concurrent chemoradiotherapy (cCHRT) (the standard treatment method for LA-HNSCC) is inappropriate due to the large tumor and expected toxicity of high-volume radiotherapy, a sequential treatment is chosen—induction chemotherapy (iCHT), followed by cCHRT, is a strong option [1,2]. Understanding the effects of iCHT on tumor biology before delivery of definitive treatment is paramount to provide as much information as possible to tailor the treatment plan to the individual patient. In the advanced stages of the disease, it makes a subsequent treatment (such as surgery, radiation therapy, or cCHRT) more effective and significantly supports organ preservation by downstaging the disease.

LA-HNSCC patients with a high risk of distant failure, multiple involved nodes, or large-volume nodal disease appear to gain certain benefits from this chemotherapy approach [3,4]. However, iCHT, like any systemic treatment, is burdened with a relatively high toxicity [5,6,7], and there is still room for improvement in its effectiveness [8,9,10]. Consequently, the debate about the impact of iCHT on locoregional and distant control, as well as overall survival, remains open [11,12,13].

Although the comparison of iCHT + cCHRT versus cCHRT alone has been studied in numerous clinical trials, revealing serious clinical implications, there is no basic research that would address more fundamental questions about molecular processes discriminating between the treatment arms. Therefore, we believe that this work has an innovative character. Two famous phase III trials, DeCIDE and PARADIGM, demonstrated that there were no significant survival differences between iCHT followed by cCHRT and definitive cCHRT [14,15]. These studies concluded that there is a marginal survival toward iCHT in patients with clinically N2c and N3 disease (bulky disease), the observation being consistent with a preliminary retrospective study which reported that survival benefits of iCHT were only for high-risk tumors (mean high-volume disease) [16].

Historically, indices of the efficacy of treatment in solid tumors have been based solely on a systematic assessment of tumor volume: changes in tumor metabolism are documented to occur early during therapy and, therefore, precede reduction in tumor volume [17].

Metabolic reprogramming is a common phenomenon that occurs in several solid tumors, among which are head and neck cancers, which allows them to survive and progress better. Among the hallmarks of cancer are a limitless replicative potential, sustained angiogenesis, avoidance of apoptosis, self-sufficiency in growth signals, insensitivity to antigrowth signals, and finally, tissue invasion and metastatic potentiality. Recently, an additional feature has also been added: reprogramming of energy metabolism [18]. Semrau et al. [19] found that the metabolic response after a first cycle of iCHT differentiates head and neck cancer patients into three subgroups that predict local tumor control. They used 18F-FDG-PET/CT based on the residual SUVmax ranges.

This work is a follow-up of our previous research, where we used NMR-based metabolomics to identify alterations in the blood serum metabolic profile due to iCHT and correlate the results with response to induction treatment [20], as well as to recognize real-time changes in the metabolome during cCHRT [21,22]. In the current study, two groups of LA-HNSCC patients undergoing cCHRT were compared: those for whom it is a first-line treatment (cCHRT) and those who have previously received iCHT (iCHT + cCHRT). Blood serum metabolic profiles have been analyzed at weekly intervals throughout the course of cCHRT, and the results have been enriched with the analysis of standard medical examination and laboratory blood parameters, as well as the nutritional and psychological statuses of the patients. This approach opens the possibility of the personalization of the monitoring and treatment of patients with HNSCC. Because the metabolic composition of blood is known to reflect the response of organisms to disease and treatment-related factors, the clinical data were supplemented with the metabolic one gathered via nuclear magnetic resonance (NMR) spectroscopy of blood serum. NMR is an analytical method—robust, reliable, highly reproducible, and noninvasive—that provides detailed structural information of organic molecules as well as enables a large number of compounds to be identified and cataloged [23]. It has been used to investigate biofluid compositions dating back to the 1980s. Moreover, it allows for monitoring of the metabolic changes due to different treatment stages or natural disease progression [24,25,26].

The gathered data are analyzed using multivariate and univariate statistical methods. Orthogonal partial least squares analysis (OPLS) was chosen to visualize how and to what extent iCHT and cCHRT duration affect the blood serum metabolic profile and clinical blood parameters. PLS (also known as partial least squares regression) or its orthogonal version, OPLS, are the methods of choice for multivariate regression analysis when dealing with multicollinear variables [27,28]. The variables affected by iCHT are further analyzed to investigate how their levels change over time during cCHRT and whether there are statistically significant differences between the iCHT + cCHRT and cCHRT groups. The Mann–Whitney U test was chosen for this task due to the non-normal distribution of the data.

The goal is to identify and describe the iCHT-induced alterations in the blood serum metabolic profiles and in the clinical blood parameters to broaden the knowledge of how induction treatment affects the tolerance of the subsequent stages of treatment.

## 2. Results

### 2.1. Influence of Induction Chemotherapy and cCHRT on the Blood Serum Metabolic Profile

An OPLS model was used to visualize the relationship between the two X variables (days from the start of cCHRT, induction/no induction) and the blood serum metabolites (Y variables). A total of 843 blood serum samples collected on a day before and during cCHRT were used. Figure 1 shows the coefficient overview plot for the OPLS model, namely, how the X variables (iCHT and duration of cCHRT) affect the blood serum metabolic profile (Y variables).

The positive/negative value indicates the increase/decrease of a metabolite level, respectively, and the error bars represent a 95% confidence interval. During the treatment with cCHRT, the patients who have previously received iCHT show higher levels of the blood serum lipids, betaine, glycine, and phosphocholine as well as lower levels of serine, glucose, tyrosine, phenylalanine, and the NMR inflammatory markers N-acetyl-glycoproteins (NAG) and N-acetlycysteine (NAC). Furthermore, along with the advance of cCHRT, we observe a strong increase of the NMR inflammatory markers (NAG and NAC) and the ketone bodies (KB; 3-hydroxybutyrate, acetoacetate, acetone) with a simultaneous decrease of phosphocholine, alanine, glutamine, carnitine, methanol, threonine, serine, glucose, and tyrosine, as well as the branched-chain amino acids (BCAA; isoleucine, leucine, valine), with the exclusion of isoleucine.

The box plots were used to visualize the time trajectories of the metabolites affected by induction chemotherapy. Figure 2 shows how the relative metabolite concentrations (integrated area under the metabolite peak) change during the cCHRT course; the lines connect the median points, and the boxes represent 25–75% percentiles. The differences between the iCHT + cCHRT and cCHRT groups were assessed using the MWU test, and the statistically significant time points are indicated with a triangle (Δ). The iCHT + cCHRT patients start cCHRT with significantly increased glycine and lipid signals, which, however, decrease during the first weeks of a therapy down to a level comparable to that detected in the cCHRT alone group (Figure 2). Similarly, the iCHT + cCHRT group enters cCHRT with a significantly higher phosphocholine. This signal also decreases strongly during cCHRT, but the significant difference between both groups persists throughout the course of cCHRT (Figure 2). The increase in betaine in iCHT + cCHRT is statistically significant at the beginning, at week 3, and at the end of cCHRT (week 7) (Figure 2). NAG and NAC are decreased in the iCHT + cCHRT group but with statistical significance only at the beginning (NAG) and during the first two weeks of cCHRT (NAG and NAC). Both NAG and NAC markedly increased during chemoradiotherapy in the iCHT + cCHRT and cCHRT alone groups (Figure 2). The iCHT + cCHRT group is also characterized by significantly decreased levels of tyrosine (at weeks 2, 4, 5, and 7), serine (at weeks 0 and 7), glucose (at week 7), and phenylalanine (at week 5) (Figure 2).

### 2.2. Influence of Induction Chemotherapy and cCHRT on Clinical Parameters

Again, an OPLS model was used to visualize the relationship between treatment-related factors (days from the start of cCHRT, induction/no induction) and clinical parameters. A total of 843 blood serum samples collected on a day before and during cCHRT were used. Figure 3 shows the coefficient overview plot for the OPLS model. The plot shows how the X variables (duration of cCHRT and induction chemotherapy) affect various clinical parameters (Y variables). A positive/negative value indicates an increase/a decrease of a given clinical parameter, respectively, and the error bars represent a 95% confidence interval.

As seen in Figure 3, induction chemotherapy results in a slightly increased nutritional performance (higher BMI and albumin (ALB)), distress, and reticulocyte count (Retic#), as well as in strongly decreased hematocrit (HCT), neutrophile, monocyte, red blood cells (RBC), hemoglobin (HGB), and platelet (PLT), and the slight decreases of C-reactive protein (CRP) and mean platelet volume (MPV) during the consecutive cCHRT treatment.

The time trajectories of the clinical parameters during the cCHRT course are shown in Figure 4; the statistically significant differences between the iCHT + cCHRT and cCHRT alone groups are indicated with a triangle.

Patients after induction treatment enter cCHRT with significantly lowered levels of HCT, CRP, neutrophil, monocyte, RBC, and HGB compared to patients without induction (Figure 4). These markedly reduced values persist for 4–6 weeks of cCHRT. PLT and MPV are slightly lowered in iCHT + cCHRT, with a periodic significance observed only for PLT (Figure 4). Retic# is significantly increased in the iCHT + cCHRT group at the beginning of cCHRT; however, after the first week of cCHRT, the levels in both groups become equal (Figure 4). Inversely, after the first week of cCHRT, there is an increase in the nutritional parameters in the iCHT + cCHRT group. Although this increase is weakly visible for BMI, it reaches a statistical significance for ALB between weeks 1 and 4 of cCHRT (Figure 4). The distress is measured on a 0-to-10 scale, and the majority of the patients do not exceed level 2. Due to the nature of this variable, the box plots are inconvenient to interpret; however, the MWU test shows that in weeks 4 and 6, the differences in the distress levels reach a statistical significance to the disadvantage of the patients after induction treatment (Figure 4).

### 2.3. Potential Biasing Factors—Tissue Volume Exposure to the Radiation Treatment

Because iCHT aims to reduce the tumor volume prior to the consecutive radio- or concurrent chemoradiotherapy, it is crucial to examine whether the tissue volumes exposed to the radiation treatment differ between the studied groups. The tissue volumes receiving the following doses (isodose volumes), 5, 10, 15, 20, 25, 30, 40, 50, 60, and 70 Gy, were compared between the cCHRT alone and iCHT + cCHRT groups, and no statistically significant differences were observed for either dose (isodose). A graphical comparison of the volumes of tissues in both groups that received a given dose of radiation is included in the Appendix A.

## 3. Discussion

Based on evidence from the literature, most patients with bulky LA-HNSCC benefit from iCHT due to the better response rate (a reduction of at least 30% in the sum of the longest diameter of the target lesions), locoregional control, progression-free survival, and distant metastatic rate [2,3,4,5]. Reduced primary tumor volume and/or partial/complete nodal remission can result in smaller volumes irradiated, making subsequent treatment less radiotoxic (smaller irradiated volume means lower acute radiation sequelae (ARS)), less adverse events (e.g., malnutrition, distress, xerostomia, dysgeusia, inflammation), and make treatment more effective (maintaining therapy schedule, fewer treatment breaks due to severe toxicity, less pharmacotherapy, better prognosis, improved organ preservation). Our previous study [21] showed that the changes in the metabolic profile of the blood serum correspond to the systemic and tumor-related responses to the therapy and are slightly dependent on the irradiated volume. The influence of the irradiated volume on metabolic profile was found to be mainly limited to sparse correlations with the inflammatory markers, creatinine, and the lymphocyte count in RT and the branched-chain amino acids in cCHRT. For the results obtained in this study to be biologically interpretable, it was necessary to check whether the irradiated volumes differed between the analyzed groups. Since the differences were not statistically significant (Appendix A), we can assume that the impact of concurrent chemoradiotherapy on the parameters analyzed is similar in both groups, and the observed differences are mainly attributable to iCHT.

### 3.1. Direct Impact of iCHT on the Metabolic and Clinical Statuses of the Patients

The results obtained in this work clearly indicate that, with few exceptions, most of the observed differences between the iCHT + cCHRT and cCHRT alone groups are due to iCHT. The patients who received the induction treatment entered cCHRT with altered metabolic and clinical profiles. The main metabolic features of their altered blood serum NMR spectra are the markedly increased lipid signals, as well as those of phosphocholine/phospholipids, glycine, and betaine (Figure 2). Previously [20], we observed a close relationship between the increase in the serum lipids and a favorable response to iCHT; however, the post-iCHT increases in phosphocholine/phospholipids, betaine, and glycine were not significant, while in the present study, the levels of these metabolites are significantly elevated in the iCHT + cCHRT group compared to the cCHRT alone group (Figure 2). The literature search indicates that an increase of phospholipids is often detected in patients favorably responding to chemotherapy [29], while a reduced concentration of betaine in the tissues is observed because of the nephrotoxic effect of cisplatin [30]—the latter reason is, thus, a possible cause of the increase in the betaine levels in the blood. Glycine is found to be increased in HNSCC patients in the tumor tissue [31] and lowered in their blood sera [32]. The literature reports on glycine do not provide a clear explanation of the relationship of this simple amino acid to the response to chemotherapy, although its increase in the blood serum after the first cycle of cisplatin-based chemotherapy [33] as well as after a surgery [32] has been observed. Glycine is involved in regulated cell death and cytoprotection [34]; on the other hand, serine/glycine biosynthesis in one-carbon metabolism is a crucial element of sustainment for cancer cell survival and proliferation [35].

As seen in Figure 2, serine, NAG, and NAC are the metabolites that decreased in the iCHT + cCHRT group at the beginning of the cCHRT course (although at week 0, the decrease of NAC is not significant). Serine is a major one-carbon unit donor during its interconversion to glycine in the folate cycle of the one-carbon metabolic network [36,37]. We can, therefore, assume that the reduced level of serine in the blood serum is due to the up-regulation of glycine production or, what seems even more likely, a down-regulated de novo serine synthesis. NAG, in turn, is a proven NMR marker of an inflammatory state [38,39], showing a very good correlation with CRP [21,22,40]. A similar role is played by NAC, which exerts a potent protective effect against oxidative stress and inflammation under different conditions [41]. It is common knowledge that advanced tumor cancers induce chronic inflammatory process, and the disease-induced inflammation is reduced by anti-cancer treatment, which we have also observed in our previous study [20]—thus, the significantly lower NAG, NAC, and CRP values in the iCHT + cCHRT group at the beginning of cCHRT compared to the patients without induction (Figure 2 and Figure 4). On the other hand, inflammation is inextricably linked to cancer treatment methods (radiation and chemotherapy), which can be easily observed in the blood serum metabolic profile [21,22,40]. Therefore, the build-up of inflammation markers during and the decrease at the end of cCHRT reflect the changes in the treatment-induced inflammation (Figure 2 and Figure 4).

As we have shown previously, iCHT results in lower NAG and CRP values in the HNSCC patients; an inflammation reduction was observed in both sexes; however, it was only statistically significant in men [20]. From the perspective of blood clinical parameters, we observe a significant decrease in CRP in the iCHT + cCHRT group commencing the cCHRT alone treatment (Figure 4), which is consistent with the above changes in the blood serum metabolic profile. Hematological toxicity is still virtually an inherent side effect of iCHT, with the most common manifestations being the reduction in red blood cells (anemia), white blood cells (neutropenia), and platelets (thrombocytopenia) [5,6,7]. After induction treatment, the patients enter cCHRT with significantly lower values of HGB, RBC, and HCT, which clearly indicates anemia [5,6,7,42], as well as significantly lower neutrophil and monocyte counts. Monocytes are also always affected by chemotherapy, but the course of these disorders is more fluctuating [43], with early monocytopenia being indicated as a risk factor for neutropenia [44]. The platelet count and the mean volume are lower at the beginning of cCHRT in the iCHT + cCHRT group but without statistical significance (Figure 4). Another post-induction difference is a significant increase in Retic# when compared to the patients without induction (Figure 4). This indicates an ongoing process of erythroid regeneration from chemotherapy-induced anemia [45].

Based on the significant changes observed in the metabolic profile of the blood serum and in the laboratory parameters of the peripheral blood, we can conclude that the patients who received iCHT start the concurrent cCHRT treatment with the following:Disturbed lipid metabolism (increased blood serum lipids and phospholipids);Deregulated folate cycle of one-carbon metabolism (elevated glycine, betaine, lowered serine);Downregulated inflammation (lowered NAG, NAC and CRP);Ongoing and/or recovery from chemotherapy-induced hematological toxicity (anemia, neutropenia, and monocytopenia).

### 3.2. iCHT Consequences during the cCHRT Course

The obtained results (Figure 2 and Figure 4) indicate that the changes induced by iCHT follow three different trends during concomitant chemoradiotherapy (cCHRT):I—iCHT-induced alterations persist throughout the whole cCHRT course;II—iCHT-induced alterations vanish during the cCHRT course;III—cCHRT enhances iCHT-induced alterations to a significant level, or iCHT intensifies cCHRT-induced alterations.

The first category (the I trend) includes only phosphocholine (Figure 2). Its levels are found to lower in both studied groups, attaining a plateau around the 4th week of the treatment; with the treatment progress, the difference in the phosphocholine levels reduces while remaining statistically significant.

Most of the analyzed metabolites and the laboratory parameters fall into the second category (the II trend). The obtained results clearly indicate that hematological toxicity (anemia, neutropenia, and monocytopenia) induced by iCHT burdens the patients during almost the entire duration of the cCHRT treatment. The differences remained statistically significant until week 6 for RBC and until week 5 for HCT, HGB, and monocyte count (Figure 4). The difference in the neutrophil counts was significant until the 4th week of cCHRT (Figure 4). Comparing the trajectories of the changes in the levels of neutrophils between the studied groups, it can be seen that after iCHT, the patients receive a neutropenia treatment a week earlier than the patients without induction (before the second week of cCHRT), and this supportive treatment is repeated after the 4th week, and from that time, the neutrophil counts equalize in both groups (Figure 4).

The difference in the blood serum lipids is significant until week 5 (the lipid signals at 0.9 and 5.3 ppm) and week 4 (the lipid signals at 1.3 ppm) of cCHRT (Figure 2). In both groups, we observe the initial decrease in the lipid levels, which plateaus in the iCHT + cCHRT group, while in the patients without induction, an upward tendency is observed—such trends cause the lipid levels to equalize in the second half of the cCHRT treatment (Figure 2). As seen in Figure 2 and Figure 4, the cCHRT treatment induces a strong inflammatory response reflected in an increase of the NMR-specific (NAG) and clinical (CRP) markers of inflammation. In the case of the NMR markers, the differences between the iCHT + cCHRT and cCHRT groups disappear quickly (in the third week of cCHRT) (Figure 2), whereas CRP remains significantly lower in the iCHT + cCHRT group until the fourth week (Figure 4), possibly due to the different half-lives between the NMR and CRP markers [38,39].

Among the metabolites involved in the folate cycle of one-carbon metabolism, the elevated glycine levels in the iCHT + cCHRT group are maintained almost throughout the cCHRT course, but the statistically significant differences occur only in the first three weeks and later in the 6th week (Figure 2). The between-group differences in betaine and serine are weaker, being significant in week 3 (betaine) and after the cCHRT completion at week 7 (betaine and serine) (Figure 2). Finally, the reticulocyte counts (Retic#), after a drastic decline after the first week of cCHRT, become equal in both groups and then do not change significantly during the further course of cCHRT (Figure 4).

The third category (the III trend) consists of the metabolites and the laboratory parameters, showing no significant differences between the groups after iCHT. Compared to NAG and CRP, NAC is the weakest marker of the course of the inflammatory processes; it is statistically significant only in weeks 1 and 2 of the cCHRT treatment (Figure 2). Comparing the trajectories of NAG, CRP, and NAC (Figure 2 and Figure 4), we can assume that the cCHRT-induced inflammatory response is stronger in the cCHRT alone group. As revealed from the studies by Huang et al. [46], concurrent chemoradiotherapy is expected to create an active immune response based on the effect induced by iCHT—this is, however, not confirmed by our observations. However, it should be kept in mind that aside from their direct impact on cancer cell mortality, the drugs used in induction therapy are also known to influence immune cell functionality and to modulate the immune cell profile in the peripheral blood [47]; thus, the net mechanism is extremely complex and requires additional multi-threaded research. NMR-based metabolomics seems to be especially valuable in this area. Our results indicate that the induction treatment weakens the immune system, leading to both weaker and delayed inflammatory reactions as compared to the group without induction. Furthermore, although NAG, NAC, and CRP are considered markers of inflammation, they reflect different aspects of the inflammatory response [38,39].

Two other metabolites in the third category are the amino acids tyrosine and phenylalanine. Tyrosine is synthesized from phenylalanine by the tetrahydrobiopterin (BH4)-dependent enzyme phenylalanine hydroxylase. Figure 2 shows that their concentrations are not significantly changed after iCHT. As revealed from the recent murine studies, the hepatic levels of phenylalanine and tyrosine may change following doxorubicin treatment, and these changes may be treated as the metabolic signatures of anthracycline-induced hepatotoxicity [48]. In turn, BH4 availability is known to be diminished under oxidative stress—the main toxic side effect of radiotherapy [49]—resulting in lowered tyrosine levels [50]. This effect seems to be more pronounced in the patients who received induction treatment. In these patients, at weeks 2, 4, and 5 of cCHRT and after the end of treatment (week 7), the tyrosine levels are significantly lower than in the patients without induction (Figure 2). On the other hand, the significant differences in the phenylalanine levels were observed only at week 5 of cCHRT (Figure 2). It can be expected that the impairment of phenylalanine hydroxylase will cause a simultaneous decrease in tyrosine and an increase in phenylalanine. However, in our case, both metabolites not only have a reduced level in the iCHT + cCHRT group but also maintain a similar trend of the changes during the cCHRT alone course (Figure 2), presumably indicating the chemotherapeutic hepatic toxicities [51]; however, it requires further studies. The last metabolite included in the third category is glucose—it is significantly reduced in the iCHT + cCHRT patients after completion of chemoradiotherapy (Figure 2). We have reported a temporary decrease in glucose levels before, in a similar group of patients, at the end of the cCHRT treatment [21], as well as a decrease in glucose correlated with the severity of ARS, although without a statistical significance [40].

In addition, two clinical parameters also fall into the III category. During cCHRT, we observe a parabolic change in the platelet counts, i.e., a decrease followed by a return to the values close to the initial ones (Figure 4). However, in the patients who received induction, the decrease in PLT was significantly stronger than in the cCHRT group, and the return to the baseline counts was slower (Figure 4). On the other hand, we do not observe any differences in the mean platelet volume. Similarly, as in chemotherapy, thrombocytopenia is a common complication during radio- and chemoradiotherapy [52]. Finally, the patients in the iCHT + cCHRT group show a higher distress at weeks 4 and 6 of cCHRT compared to those without iCHT (Figure 4). The depressive symptoms are very common in patients with HNSCC undergoing radiotherapy [53], and our results indicate that induction chemotherapy before cCHRT may be an additional intensifying stimulus.

The above results indicate that

Most of the metabolic and clinical alterations induced by iCHT are temporary and disappear during or at the completion of cCHRT;The only lasting effect, maintained until at least a week after the end of cCHRT, is a significantly elevated level of phosphocholine;iCHT may intensify some of the metabolic and clinical alterations caused by cCHRT (phenylalanine—tyrosine metabolism; thrombocytopenia; distress).

What needs to be stressed is that the effectiveness of the subsequent treatment, i.e., radio- or chemoradiotherapy, depends mainly on administering the full planned dose at the planned time. Any interruption of treatment negatively affects the likelihood of recovery. On the other hand, the probability of a cure increases as the intensity of treatment increases. The results of our publication indicate that patients after iCHT start cCHRT with an altered metabolic profile and clinical parameters (indicating hematologic toxicities), but this does not affect the tolerability of cCHRT (in the study group, patients with iCHT undergo cCHRT just like those without iCHT, and they do not have interruptions in treatment or other life-threatening events). Moreover, most induction-induced changes disappear after cCHRT.

### 3.3. Limitations of the Study

Among this study’s limitations, there are some issues that need to be addressed. First, iCHT was implemented using two different regimens (TPF and PF). We compared the analyzed parameters in the context of the iCHT regimen; no significant differences (MWU test) were observed, except for distress, which was significantly higher in the patients treated with PF. Secondly, because head and neck cancers are more common in men, there are three times more men than women in the study group. Therefore, the presented results are mainly representative of the male population. However, from the point of view of the analyses performed, the most important thing is that the difference in the number of women and men in the cCHRT and iCHRT + cCHRT groups is not statistically significant. It can, therefore, be assumed that gender has a negligible impact on the observed differences between the groups. The age difference between the studied groups is statistically significant. However, our aim was to address the real clinical situation in which there will always be an age difference between the patients who receive and those who do not receive induction chemotherapy (it is well known (without EBM) that older age is one of the indirect factors that can influence withdrawal from iCHT due to the higher risk of comorbidities and lower performance status). However, we tried to ensure that the age difference in the study groups was not too large. In the induction group, there are only three patients below the minimal age for the no-induction group (i.e., one younger than 41 years, one of them is 22, one 34, and one 40 years old).

## 4. Materials and Methods

### 4.1. Characteristics of the Patients’ Groups

This retrospective study was approved by the Ethics Committee and informed written consent was obtained from the participants. The studied group consisted of 107 HNSCC patients, 81 men and 26 women, all Caucasians, between 22 and 75 years (median 57 years) and treated in the 1st Department of Radiation and Clinical Oncology of the Maria Sklodowska-Curie National Cancer Research Institute, Gliwice Branch, Poland. All were pathologically confirmed with squamous cell carcinoma according to the third edition WHO scale. A total of 104 patients had clinical (TNM) stage III to IVB, according to the seventh edition of the American Joint Committee on Cancer (AJCC). Three patients had stage II, according to the 7th edition of AJCC, with poor prognostic factors such as tumor hypoxia, active smoking, long-term diagnostic procedures, tumor fragmentation during excision biopsy, difficulties with histopathology diagnosis (2 or more excision biopsies), and rapid progression during diagnostic procedures. The patients aged 20 years or older with primary tumors of the oropharynx, hypopharynx, nasopharynx, and larynx were included for the analysis. The patients had biopsy-proven squamous cell carcinoma in the primary and/or cervical lymph nodes. All participants had a performance status of less than 2 (ZUBROD WHO scale), good nutritional status, and were inoperable or suitable for organ preservation strategy. The exclusion criteria were as follows: distant metastases; squamous cell carcinoma of the nose, sinus, ear, skin, thyroid, or salivary glands; patients with a previous secondary malignancy; a performance status greater than or equal to 2 (ZUBROD WHO scale); or no permission for chemotherapy.

All patients were treated with radical intent by cCHRT. All of them received cCHRT using a conventional fractionation technique (2 Gy per fraction, 35 fractions, total dose 70 Gy, delivered once a day and 5 days a week with a weekend break for 7 weeks). Cisplatin was fractionated from the first day of RT, with 21-day interval (2–3 cycles). A total of 54 patients received iCHT prior to cCHRT (iCHT + cCHRT) because of a high risk of distant failure, multiple involved nodes, large volume of the primary tumor, and large volume of nodal disease. The iCHT patients were treated with docetaxel/cisplatin/5-fluorouracil (TPF) (26 patients) and cisplatin/5-fluorouracil (PF) (28 patients). The patients who responded to iCHT (a reduction of at least 30% in the sum of the longest diameter of the target lesions) received cCHRT. The detailed patients’ characteristics are presented in Table 1.

### 4.2. Clinical Monitoring of Patients

The clinical monitoring was performed weekly from the day before cCHRT (week 0) to the week after the cCHRT completion (week 7; a total of 8 time points for each patient). It included the medical, psychological, and endoscopic examinations; the laboratory blood tests; as well as the 1H NMR spectroscopy of blood serum [20,21,22]. The medical and endoscopic examinations were used to evaluate tumor regression and grade of acute radiation sequelae (ARS) [54]. The psychological examination was used to evaluate the distress (using Distress Thermometer [55]) and pain levels (using Numerical Pain Rating Scale [56]). The laboratory blood test was used to evaluate the hematological and nutritional statuses, the renal and hepatic functions, and the inflammation reaction.

### 4.3. Blood Serum Samples Collection

The peripheral venous blood samples were collected once a week (from week 0 to week 7 of cCHRT, after the patient fasted overnight). A total number of 856 samples (107 patients; 8 time points for each patient) were collected; however, 13 samples were excluded due to insufficient amount of material. The 843 samples were divided into two aliquots, one of which was used for the laboratory blood tests, and the other was used for the 1H NMR spectroscopy. The samples were incubated for 30 min at room temperature and then centrifuged (1000× *g*, 10 min) to remove a clot and stored frozen at −80 °C until NMR measurements were performed.

### 4.4. Serum Sample Preparation for NMR 

The serum samples were thawed in two steps (at 4 °C and at room temperature) and mixed with the phosphate buffer (pH 7.4) containing deuterium oxide (D2O) and trimethylsilylpropionate (TSP). The aliquots of 600 μL of the solution were transferred into 5 mm Wilmad WG-1235-7 NMR tubes (Wilmad Labglass, Vineland, NJ, USA) and kept at 4 °C until the NMR analysis.

### 4.5. NMR Measurement Protocol

The same measurement protocol (based on the protocol proposed by Beckonert et al. [23]) as in our previous metabolomic studies [20,21,22,40] was applied. The 1H NMR spectra were acquired on a Bruker 400 MHz Avance III spectrometer (Bruker Biospin, Rheinstetten, Germany) equipped with a 5 mm PABBI probe. The quality control tests were performed on every measurement day. The NMR probe tuning and matching, shimming, determination of the transmitter offset value for the water pulse presaturation, and 90° pulse adjustments were always made for each sample. The receiver gain was set to 90.5, and the temperature was set to 310 K for all experiments. Four different 1H NMR spectra—NOESY (1D nuclear Overhauser enhancement spectroscopy), CPMG (Carr–Purcell–Meiboom–Gill), diffusion-edited (DIFF), and J-resolved (JRES)—were acquired for each serum sample. The characteristics of the acquired spectra, as well as the pulse sequence parameters, are provided in Appendix A.

### 4.6. Spectra Post-Processing 

One-dimensional spectra were processed with a line broadening of 0.3 Hz and automatically phase-corrected (in Topspin software version 3.1 from Bruker Biospin), referencing the methyl doublet of alanine at 1.5 ppm and bucketed over the region of 9.0–0.5 ppm with the bucket width set to 0.002 ppm using AMIX software version 4.0.2 (Bruker Biospin). The spectrum region of water (5.15–4.38 ppm, d = 0.77 ppm) was removed from the analysis in order to prevent variation in each sample. No normalization was applied. This is the standard processing protocol used in our metabolomic lab [20,21,22,40].

### 4.7. Metabolite Identification and Quantification

The identification of the metabolites was carried out based on the comparisons with the reference compounds library (in Chenomx NMR Suite Professional (Chenomx Inc., Edmonton, AB, Canada)), as well as on the multiplicity and scalar couplings information extracted from the 2D JRES spectra, and using the information from Human Metabolome Database (http://www.hmdb.ca/ accessed on 18 October 2023) and the available literature.

The low-molecular-weight metabolites were quantified based on the 1D positive projections of the JRES spectra. The diffusion-edited spectra were used for quantification of the lipid signals. The integrals were measured in the spectral regions defined individually for each metabolite using the “sum all points in region” method in AMIX (Bruker Biospin) software.

### 4.8. Statistical Analysis

The multivariate analyses were carried out on the 843 samples using SIMCA (Sartorius Stedim Data Analytics, v. 17, Umeå, Sweden) software. Each sample was described by 62 descriptive variables (47 NMR metabolites and 15 nutritional, psychological, and standard laboratory blood parameters) and 2 property variables (iCHT: 0–1 variable; cCHRT: days after starting the treatment, a continuous variable). The descriptive variables were scaled to unit variance (UV) and mean-centered prior to modeling. Orthogonal partial least squares analysis (OPLS) was used to identify which descriptive variables are altered by property variables and to what extent. OPLS combines an orthogonal signal correction with partial least squares (also known as projection to latent structures (PLS)) analysis; a detailed description of OPLS is available in [27,28]. In short, orthogonal signal correction removes a variation from the descriptive variables (NMR metabolites and clinical parameters) that is not correlated to the property variables (iCHT and cCHRT duration). Then, PLS is performed on the filtered data. The objective of PLS is to visualize the relationship between the descriptive and property variables. The so-called coefficient overview plots (Figure 1 and Figure 3) are then used to visualize the influence of iCHT and cCHRT duration on the analyzed descriptive variables. The OPLS coefficients are a dimensionless measure, usually normalized to the range [−1, 1] to make them comparable when the property variables have different ranges. The coefficient overview plot displays the coefficients for all property variables as the vertical bars. The height of the bar determines how strongly the analyzed property variable affects a given descriptive variable, and the positive and negative values determine the increase or decrease in the value of a given variable under the influence of the analyzed parameter. The error whiskers represent 95% confidence interval.

The univariate statistical analyses, e.g., the Mann–Whitney U test (MWU), were carried out using Statistica software version 12 (Statsoft, Tulsa, OK, USA). The significance threshold was set at 0.05. The MWU test was used to check whether the differences observed between the groups (in individual weeks of cCHRT) were statistically significant (Figure 2 and Figure 4).

## 5. Conclusions

In summary, this retrospective study shows that the bulky LA-HNSCC patients who received induction chemotherapy begin consecutive treatment (in this case, concurrent chemoradiotherapy) with significant alterations in the metabolic and clinical profiles. iCHT results in disturbed lipid and one-carbon metabolism, downregulated inflammation, and hematological toxicity. However, these burdens are found to normalize during cCHRT, with the exception of betaine, serine, tyrosine, glucose, phosphocholine, and the platelet count, where the between-group difference remains constant or decreases but remains significant.

The main conclusion from the above study is that in the bulky LA-HNSCC patients treated with radical intent, the use of induction chemotherapy results in increased toxicity at the beginning of cCHRT, but ultimately, the final toxicity of iCHT + cCHRT vs. cCHRT alone is comparable from a certain stage of the treatment, and the patients with and without iCHT tolerate cCHRT similarly. This means that the use of iCHT improves treatment conditions in advanced cases without the risk of exacerbating the overall toxicity compared to cCHRT alone.

## Figures and Tables

**Figure 1 ijms-25-00188-f001:**
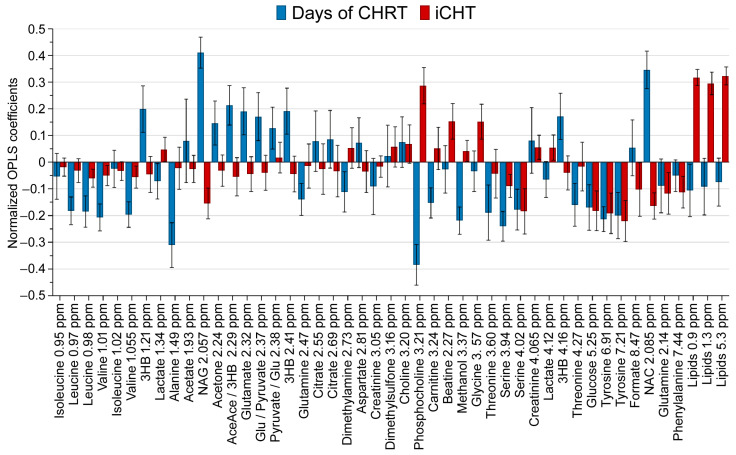
OPLS coefficient overview plot showing how iCHT and the cCHRT duration affect the blood serum metabolites. A total of 843 blood serum samples collected on a day before and during cCHRT were used in the model. The OPLS coefficients are calculated for the two analyzed property variables (duration of CHRT in days—a continuous variable; iCHT—a discrete variable (1 or 0)) normalized to the range [−1, 1] and displayed as the vertical bars. The height of the bar determines how strongly the analyzed property variable (iCHT, cCHRT duration) affects a given descriptive variable (blood serum metabolite), and the positive and negative values determine the increase or decrease in the value of a given metabolite under the influence of the analyzed property variable. The error whiskers represent 95% confidence interval. The results take into account the resonance signals of metabolites at different chemical shifts, e.g., isoleucine at 0.95 and 1.02 ppm. Legend: 3HB—3-hydroxybutyrate; NAG—N-acetyl-glycoprotein; AceAce—acetoacetate; Glu—glutamate; NAC—N-acetylcysteine.

**Figure 2 ijms-25-00188-f002:**
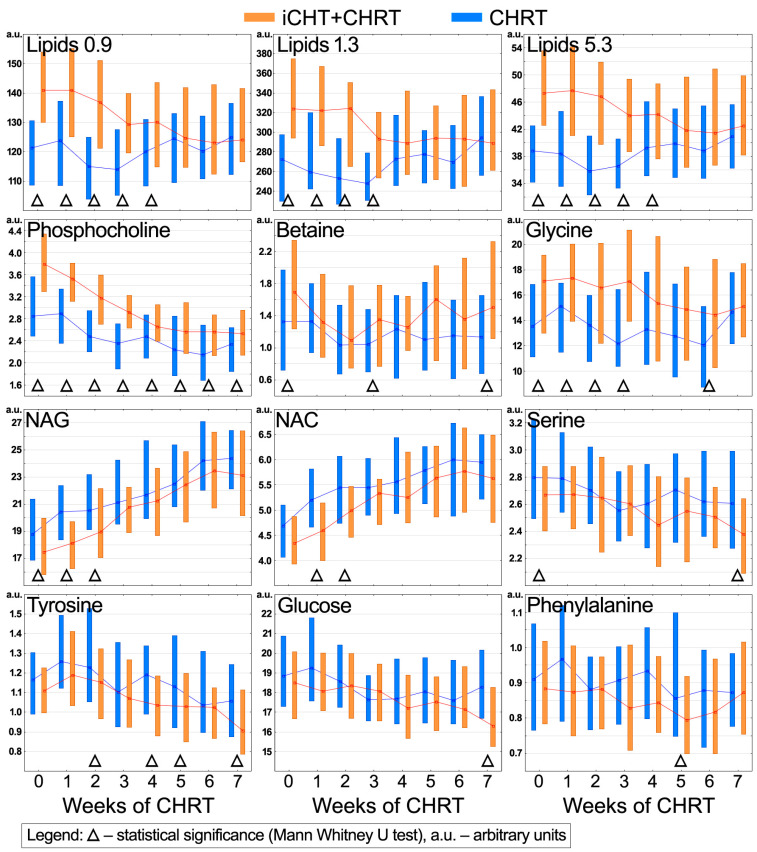
The box plot representation of the relative changes during the course of cCHRT in the metabolites identified by OPLS as significantly affected by the induction treatment. The lines connect the median points, and the boxes represent 25–75% percentiles. The differences between the iCHT + cCHRT and cCHRT alone groups were assessed using the MWU test, and the statistically significant time points are indicated with a triangle (Δ). The relative concentrations of the metabolites are provided in arbitrary units (a.u.) that do not correspond to the physiological values.

**Figure 3 ijms-25-00188-f003:**
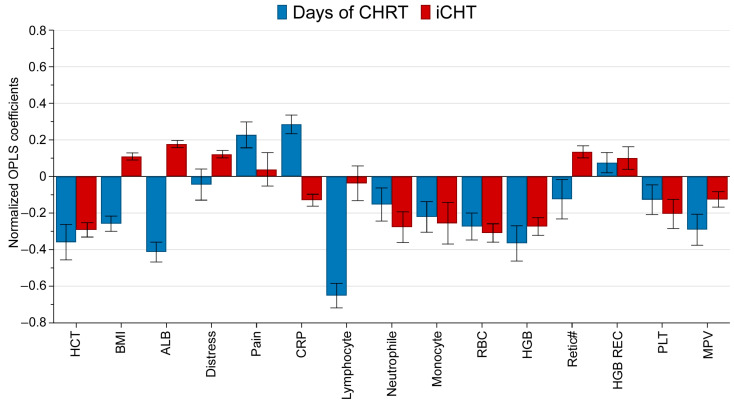
OPLS coefficient overview plot showing how the duration of cCHRT and iCHT affect the clinical parameters. A total of 843 blood serum samples collected on a day before and during cCHRT were used in the model. The OPLS coefficients are calculated for the two analyzed property variables (duration of CHRT in days—a continuous variable; iCHT—a discrete variable (1 or 0)) normalized to the range [−1, 1] and displayed as the vertical bars. The height of the bar determines how strongly the analyzed property variable (iCHT, cCHRT duration) affects a given descriptive variable (clinical parameter), and the positive/negative values determine the increase/decrease in the value of a given clinical parameter under the influence of the analyzed property variable. The error whiskers represent 95% confidence interval. HCT—hematocrit; ALB—albumin; CRP—C-reactive protein; RBC—red blood cell count; HGB—hemoglobin; Retic#—reticulocyte count; HGB REC—hemoglobin concentration in reticulocytes; PLT—platelet; MPV—mean platelet volume.

**Figure 4 ijms-25-00188-f004:**
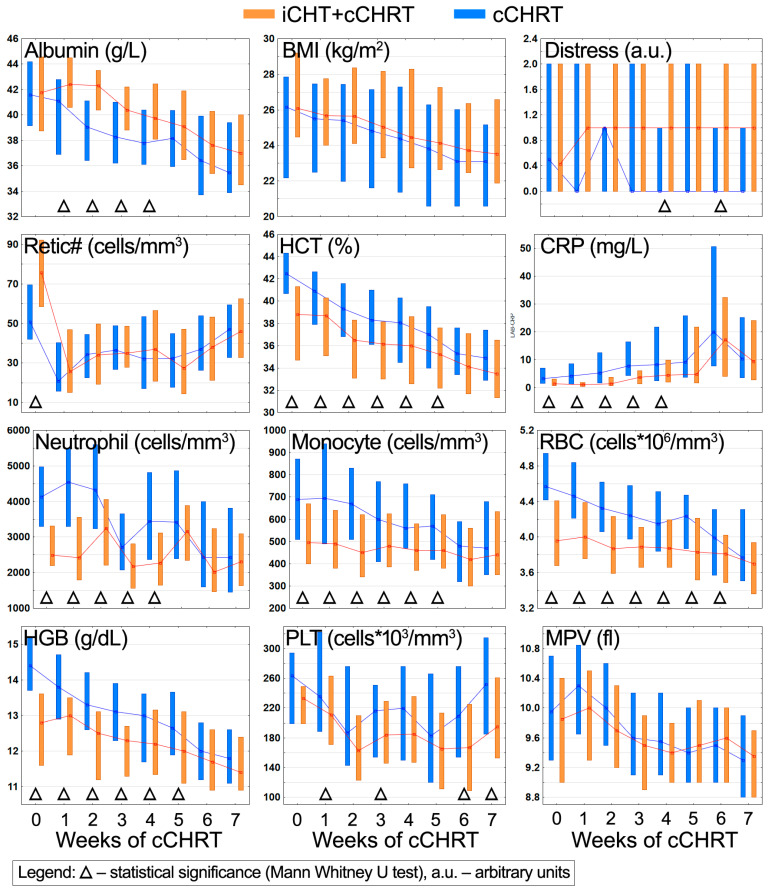
The box plot representation of the relative changes during the cCHRT course in the clinical parameters identified by OPLS as significantly affected by the induction treatment. The lines connect the median points, and the boxes represent 25–75% percentiles. The differences between the iCHT + cCHRT and cCHRT groups were assessed using the MWU test, and the statistically significant time points are indicated with a triangle. Retic#—reticulocyte count; HCT—hematocrit; CRP—C-reactive protein; RBC—red blood cell count; HGB—hemoglobin; PLT—platelet; MPV—mean platelet volume.

**Table 1 ijms-25-00188-t001:** Characteristics of the study group.

	cCHRT	iCHT + cCHRT
Number of patients	53	54
Age (median)	60 (41–75)	55 (22–71)
Sex		
M	43	38
F	10	16
TNM stage		
II	3	
III	12	10
IV	38	44
Tumor localization		
Larynx	22	12
Oropharynx	15	18
Hypopharynx	13	11
Nasopharynx	3	13
HPV status		
Positive	1	0
Negative	52	54

## Data Availability

The datasets generated and/or analyzed during the current study are available from the corresponding author on a reasonable request.

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
