# Peer review of "Metabolomic Insight into Implications of Induction Chemotherapy Followed by Concomitant Chemoradiotherapy in Locally Advanced Head and Neck Cancer"

_ijms, 2023, doi:10.3390/ijms25010188_

Round 1

Reviewer 1 Report

Comments and Suggestions for Authors

The synergy of cancer therapies remains crucial due to the emergence of new and enhanced treatment outcomes. Patient data obtained could also benefit the formulation of other treatment plans. However, certain aspects need clarification before this manuscript can be published.

The English language used in the text is challenging to comprehend. Expressions like 'statistical importance,' 'iCHT was realized with docetaxel-cisplatin and 5-fluorouracil (TPF),' and 'The overnight fasting blood samples from the peripheral vein were collected weekly on each day of the ARS evaluation' make it difficult to follow.

I'm uncertain about the novelty presented in this draft. While iCHT and cCHRT are well-known and extensively documented, perhaps their application in head and neck squamous cell carcinomas is inadequately researched.

Why there is no name for the Y axes in Fig 1? What are these 1,2 numbers in the x-axis in the name? I would rearrange Fig. 1 in a more understandable way. At what time they were measured? Aren’t the same data presented in the other few graphs?

What are the values presented in Fig 2? I understand that they are not equal, but then one can normalize them to appropriate numbers like 1 or percentages. Therefore, it is illogical to present values in a graph that are not presented in a numerical form. Also, what do these triangles mean? Statistical significance? Please write the meaning in the captions.

Also please name the y-axis in Fig 3 the same as in Fig 1. At what time the values were taken? Also, is the data presented in Fig 3 also presented in Fig 4? In Fig 4 y axis numbers already appaired. In Fig 4 please write what triangles mean in the graph.

In the discussion part, I miss the survival rate of the patients or at least the values of decreased tumours or something like this. Without them, it is just blood value change. Of course, it is clear it will change since the body is being intoxicated and affected by ionizing radiation. To my understanding no novelty there.

In methodology, table 1 is not clear. What is II, III, or IV (probably stages of cancer? Or maybe the age groups?).

How was the psychological examination done (distress and pain levels)?

In conclusion, the authors state about successive treatment. In which way it was evaluated where is the data about this? The reviewer only can find blood serum and blood cell changes.

Comments on the Quality of English Language

The English language used in the text is challenging to comprehend. Expressions like 'statistical importance,' 'iCHT was realized with docetaxel-cisplatin and 5-fluorouracil (TPF),' and 'The overnight fasting blood samples from the peripheral vein were collected weekly on each day of the ARS evaluation' make it difficult to follow.

Reviewer 2 Report

Comments and Suggestions for Authors

In this study, Boguszewicz et al. investigated the impact of induction chemotherapy (iCHT) on blood serum metabolic profiles and clinical parameters in locally advanced head and neck squamous cell carcinoma (LA-HNSCC) patients undergoing concurrent chemoradiotherapy (cCHRT). They found transient disturbances in lipid and one-carbon metabolism, downregulated inflammation, and hematological toxicity, but noted that these complications were temporary and mostly resolved by the end of treatment. The study suggests that iCHT prior to cCHRT may not represent a significant burden and could be considered as a safe treatment option for LA-HNSCC.

It is a good work, however, needs some improvements:

1) The current version of abstract is extensive. Kindly reduce it.

2) Details on NMR is not very informative. Kindly provide more information.

3) Provide a Graphical abstract

4) Why did authors use Orthogonal partial least squares analysis (OPLS) and Mann-Whit- 27 ney U test (MWU), provide brief information on it in Introduction.

5) Figure 2 lacks sufficient information like Y axis values, kindly revise it.

6) Add a brief note on NMR in Introduction section.

Comments on the Quality of English Language

Minor editing of English language required

Round 2

Reviewer 1 Report

Comments and Suggestions for Authors

As I did not receive a reply from the authors (only the manuscript), I only presume the changes in from the manuscript. Admittedly much has been changed. Text and the graphs are more understandable, however, some of the concerns still remain.

I'm uncertain about the novelty presented in this draft. While iCHT and cCHRT are well-known and extensively documented, even by the same authors. Can the authors comment on what is novel here?

Regarding fig.1 Why there is 2 Leucines, 2 Isoleucines, and so on? What is Days of CHRT in Legend? Perhaps it's only CHRT? Aren’t the same data presented in the other few graphs? If at least partly yes, then could I hear the comment from the authors on why the data is duplicated?

In the Introduction authors state “The goal is to identify and describe the iCHT-induced alterations in the blood serum metabolic profiles and in the clinical blood parameters to broaden the knowledge of how induction treatment affects the tolerance of the subsequent stages of treatment”. Indeed, the results answer this. However, in conclusion, the authors state “this retrospective study shows that the bulky LA-HNSCC patients who received induction chemotherapy begin successive treatment (in this case the concurrent

chemoradiotherapy) with the significant alterations in the metabolic and clinical profiles.” From the presented data in this manuscript (not in other data that is published in a separate publication), one can NOT presume this. This is not a discussion that one can discuss. This is the conclusion. Therefore, the conclusions cannot be drawn from the presented results.

Author Response

In case there are problems with the attachment, we paste the content of the responses to the first and second round of reviews below.

--------------------------------------------------------------------------------

REVIEWER 1 – round 2

Dear Reviewer, We are very sorry that you did not receive our answers to your questions. Everything indicates that this was some kind of error in the MDPI system, as it is not possible to return a revised manuscript without first adding a response to the reviewers. As evidence, we include the responses from the first round of reviews at the end of this document. Since some of the questions are still the same, the answers in the rounds 1 and 2 may overlap.

Answers to the questions from the second round of the reviews

Comment 1

I'm uncertain about the novelty presented in this draft. While iCHT and cCHRT are well-known and extensively documented, even by the same authors. Can the authors comment on what is novel here?

Answer

iCHT and cCHRT are well-known treatment methods in the advanced stages of head and neck squamous cell carcinoma. Evidence-based medicine, e.g. NCCN guideline data (National Comprehensive Cancer Network 1.2024) shows that the preferred chemoradiotherapy approach for fit patients with locally advanced disease remains concurrent cisplatin and radiation therapy. Cisplatin-based induction chemotherapy can be used, followed by a radiation-based locoregional treatment (i.e., sequential cCHRT). However, an improvement in an overall survival with the incorporation of induction chemotherapy compared to the direct introduction of cCHRT) has not been established in the randomized studies. Cisplatin-based induction chemotherapy followed by high-dose, every 3 weeks, cisplatin chemoradiotherapy is associated with toxicity concerns. After iCHT, multiple options can be used for the radiation-based portion of therapy, particularly for the patients with a complete remission after iCHT.

The question is who benefits from iCHT. To answer this question, first, the goal of iCHT therapy must be outlined – it is twofold: to achieve the highest possible degree of regression with toxicity that does not lead to serious complications.. There are no evidence-based data to select good responders who have no serious toxicity. Today, iCHT is an option for the patients with fast or bulky disease progression with the risk of distant spread. In our study the NMR and MS based metabolomics has been shown as the appropriate tool to detect and analyze the metabolomic changes after induction treatment in terms of its toxicity in two groups of the patients: the 'state of the art' group, cCHRT, and the 'out of the state of the art' group, iCHT followed by cCHRT. The main idea of such a combination of the groups is a pure evaluation of iCHT toxicity.

According to our review of the literature on the treatment of head and neck cancer, which we have been conducting for several years, the toxicity of iCHT is well documented, but we have not found any publication that directly compares the patients with and without iCHT during the subsequent stages of treatment in terms of the toxicity and tolerability of treatment. This is definitely new and that is why we stressed this point in the introduction of the paper.

The effectiveness of the subsequent  treatment, i.e. radio- or chemoradiotherapy, depends mainly on administering a full planned dose at the planned time. Any treatment interruption negatively affects the probability of recovery. On the other hand, the probability of cure increases as the intensity of the treatment increases. Until now, the patients after iCHT were believed to be at risk and that subsequent treatment should be carefully intensified. The results of our publication indicate that, yes, the patients after iCHT start chemoradiotherapy (cCHRT) with an altered metabolic profile and the clinical parameters (indicating hematologic toxicities), but this does not affect the tolerability of cCHRT (in the study group, the patients undergo cCHRT just like those without iCHT, they do not have any treatment interruptions or other life-threatening events). Moreover, most induction-induced changes disappear after cCHRT. This is another novelty that has been included in the discussion.

These results clearly indicate that the concerns about the use of iCHT itself (due to toxicity and uncertain response rate) and the intensity of the subsequent stages of treatment are not justified.

 Comment 2a

Regarding fig.1 Why there is 2 Leucines, 2 Isoleucines, and so on?

Answer

Nuclear magnetic resonance works by exciting the measured sample (placed in a magnetic field) with a radio frequency pulse. In the case of proton resonance, the frequency of this pulse is appropriate for the resonant frequency of protons in a given magnetic field (for 9.4 T it is about 400 MHz). In case of complex substances (blood serum in particular), electron density around the proton results in shielding/deshielding effects, so different types of protons (aliphatic, aromatic or aldehydic) present in different ranges of chemical shifts (at different places at the ppm scale) – thus, the various chemical groups of a given chemical compound may experience slightly different magnetic fields. As a result, the measured resonance signal appears in the form of several peaks with different chemical shifts.

An explanation of this phenomenon can be found in materials on the physical basics of NMR:

https://www.acdlabs.com/blog/the-basics-of-interpreting-a-proton-nmr-spectrum/

https://kpu.pressbooks.pub/organicchemistry/chapter/6-6-1h-nmr-spectra-and-interpretation/

In Human Metabolome Database one can take a look at the NMR spectra of different metabolites, e.g. leucine:

https://hmdb.ca/spectra/nmr_one_d/99907

The spectrum of blood serum is a superposition of the signals from all metabolites detectable using the NMR technique and is very complex, which is clearly visible in Figure 1 in the publication below:

Psychogios N, Hau DD, Peng J, et al. The human serum metabolome. PLoS One. 2011;6(2):e16957. Published 2011 Feb 16. doi:10.1371/journal.pone.0016957

https://www.ncbi.nlm.nih.gov/pmc/articles/PMC3040193/

We realize that the above explanation may lead to another question: why describe several signals of one metabolite, isn't one signal enough?

By analyzing all signals coming from a given metabolite, we are sure that we have made the correct identification. If two signals from the same metabolite are found to respond differently to the same stimulus, this means that the identification may be incorrect or the signal overlaps with the signal of another metabolite. Therefore, in order to provide the reader an opportunity to verify and validate the methodology applied in the publication, the analysis of the full spectra with all the NMR lines for all metabolites are necessary. We make an exception for glucose, which has a dozen or so signals in the proton NMR spectrum, most of which overlap with the signals from other metabolites. Therefore, for glucose, we focus on the signal at 5.25 ppm due to the smallest quantification error. When it is impossible to separate individual peaks in the spectrum, we note that the analyzed signal comes from two metabolites, e.g. AceAce / 3HB (Acetoacetate / 3-hydroxybutyrate).

Additional explanation is now added to the caption of Figure 1 (marked in green).

 Comment 2b

What is Days of CHRT in Legend? Perhaps it's only CHRT?

The OPLS model which results are presented in figures 1 and 3

Answer

The OPLS model, the results of which are presented in Figures 1 and 3, analyses the relationship between two data matrices:

- the matrix of the descriptive variables – the NMR metabolites (Fig. 1) or the clinical blood serum parameters (Fig. 3)

- the matrix of the property variables – a duration of CHRT in days ("Days of CHRT" - a continuous variable) and iCHT - a discrete variable (yes = 1 or no = 0).

Thus Days of CHRT is ranged between 0 to the last day of CHRT (approximately 50 – 7 weeks of CHRT) and the blue vertical bars in the figures 1 and 3 display how the DURATION of CHRT (and the increasing RT dose, because the dose is accumulated after each fraction) affects the analyzed parameter.

This is particularly visible in Figure 3. Lymphocytes are very sensitive to ionizing radiation and as CHRT progresses (the number of days increase), we observe a strong decline in their number.

Figures 1 and 3 and Figures 2 and 4 show two similar time-lines. Figures 1 and 3 show the results, where CHRT progress is measured in days. Figures 2 and 4 measure the progress in weeks. This is due to the fact that CHRT is carried out over 7 weeks (with weekend breaks) and in our opinion, such a presentation makes it easier to track the progress of changes.

 Comment 2c

Aren’t the same data presented in the other few graphs? If at least partly yes, then could I hear the comment from the authors on why the data is duplicated?

Answer

We provided the answer to this question in the previous round of the reviews, which for the reasons unknown to us, you did not receive. We will of course provide further clarifications as appropriate, but we also encourage you to read the previous responses at the bottom of this document.

Figures 2 and 4 are an extension of the results presented in Figures 1 and 3. This is not a duplicate.

The OPLS analysis, the results of which are presented in Figures 1 and 3, was performed on the entire set of the measured and determined NMR metabolites and the clinical parameters from the blood serum. The aim was to identify the variables wchich levels changed immediately after iCHT, but also during CHRT as a cumulative effect after iCHT.

However, what we see in Figures 1 and 3 does not answer the following questions:

- do the observed changes appear immediately after iCHT or do they appear during CHRT as a result of a cumulative toxicity?

- what are the differences during and at the end of CHRT between the patients who did or did not receive iCHT?

- are these differences statistically significant?

The answers to the above questions were the key to formulating the conclusions in this work.

These answers were obtained through the subsequent analyses. First, based on the results obtained from OPLS (Figures 1 and 3), we selected the variables that changed under the influence of iCHT. We then presented trajectories of change using box plots and by treatment modality. These charts clearly show the nature of changes in the analyzed parameters during CHRT and at the end of treatment. Furthermore, we assessed the statistical significance of the observed differences between both groups.

Again, Figures 1 and 2 (and 3 and 4) complement each other, not duplicate each other.

Comment 3

In the Introduction authors state “The goal is to identify and describe the iCHT-induced alterations in the blood serum metabolic profiles and in the clinical blood parameters to broaden the knowledge of how induction treatment affects the tolerance of the subsequent stages of treatment”. Indeed, the results answer this. However, in conclusion, the authors state “this retrospective study shows that the bulky LA-HNSCC patients who received induction chemotherapy begin successive treatment (in this case the concurrent chemoradiotherapy) with the significant alterations in the metabolic and clinical profiles.” From the presented data in this manuscript (not in other data that is published in a separate publication), one can NOT presume this. This is not a discussion that one can discuss. This is the conclusion. Therefore, the conclusions cannot be drawn from the presented results.

 Answer

To answer this question precisely, we need to address two issues (one of which has already been explained in the previous round of the reviews).

First of all, as "successive", we meant the consecutive stage of the treatment. In no case do we refer to the assessment of treatment success. To avoid such ambiguities, we have replaced the word "successive" with "consecutive" in the conclusions.

Secondly, the patients after iCHT DO start the consecutive treatment with the significant alterations in the metabolic and clinical profiles. This is seen in Figures 2 and 4 e.g. for serum lipids, neutrophil, HGB, etc.

Final comments from the authors

We hope that the explanations presented will allow for a positive re-evaluation of our work. For additional explanations, please read our previous responses (below), in which we have made every effort to respond to all allegations. We have no idea why you didn't receive these replies, it certainly wasn't our fault.

At the same time, thank you once again for spending so much time and preparing such an insightful review. Thank you for all your critical comments and interesting questions that allowed us to significantly improve our publication.

REVIEWER 1 – round 1

The synergy of cancer therapies remains crucial due to the emergence of new and enhanced treatment outcomes. Patient data obtained could also benefit the formulation of other treatment plans. However, certain aspects need clarification before this manuscript can be published.

The English language used in the text is challenging to comprehend. Expressions such as'statistical importance,' 'iCHT was realised with docetaxel-cisplatin and 5-fluorouracil (TPF),' and 'The overnight fasting blood samples from the peripheral vein were collected weekly on each day of the ARS evaluation' make it difficult to follow.

 Dear Reviewer, thank you for the review and the valuable comments that helped us to significantly improve our manuscript. We addressed all technical and linguistic issues and explained in detail why this is an innovative study and what the main idea of the research hypothesis is. We hope that the explanations provided will allow for a positive reassessment of the submitted manuscript.

 I'm uncertain about the novelty presented in this draft. While iCHT and cCHRT are well-known and extensively documented, perhaps their application in head and neck squamous cell carcinomas is inadequately researched.

 iCHT and cCHRT are well-known treatment methods in the advanced stages of head and neck squamous cell carcinoma. Evidence-based medicine, e.g. NCCN guideline data (National Comprehensive Cancer Network 1.2024) shows that the preferred chemoradiotherapy approach for fit patients with locally advanced disease remains concurrent cisplatin and radiation therapy. Cisplatin-based induction chemotherapy can be used, followed by radiation-based locoregional treatment (ie, sequential cCHRT). However, an improvement in overall survival with the incorporation of induction chemotherapy compared to the direct introduction of cCHRT) has not been established in randomised studies. Cisplatin-based induction chemotherapy followed by high-dose, every 3 weeks, cisplatin chemoradiotherapy is associated with toxicity concerns. After iCHT, multiple options can be used for the radiation-based portion of therapy, particularly for patients with complete remission after iCHT.

 The question is who benefits from iCHT. This goal has two parts: the first is a regression rate, and the second is toxicity without serious complications. There are no evidence-based data to select good responders who have no serious toxicity. Today, iCHT is an option for patients with fast or bulky disease progression with the risk of distant spread. The study analyses the metabolomic changes after induction treatment in terms of its toxicity in two groups 'state of the art' group, cCHRT, and 'out of the state of the art' group, iCHT followed by cCHRT. The idea is a pure evaluation of iCHT toxicity.

Why there is no name for the Y axes in Fig 1? What are these 1,2 numbers in the x-axis in the name? I would rearrange Fig. 1 in a more understandable way. At what time they were measured? Aren’t the same data presented in the other few graphs?

Answer 1.

According to the suggestion we improved Fig 1. The description of the coordinate Y-axis has been added. The numbers after the metabolite name denoted different signals of a particular metabolite observed at different positions of the ppm scale. These numbers are now replaced by the exact ppm position of a particular signal. The OPLS model was constructed using the whole pool of the collected samples (843 blood serum samples collected on a day before and during cCHRT) – it is now clearly indicated in the results section.

The OPLS model was used to identify the affected metabolites, we do not differentiate the patients here based on the treatment they received (Fig 1 and 3): both show the coefficients overview plot for the OPLS model, however, Figure 1presents how the X variables (iCHT and duration of cCHRT) affect the blood serum metabolic profile (Y variables), whereas Figure 3 shows how the X variables (duration of cCHRT and iCHT) affect various clinical parameters (Y variables).

In turn, the box plots (Fig 2 and 4) compare the changes in the metabolites and clinical parameters levels occurring over time and by the treatment received. Figure 2 shows how the relative metabolite concentrations (integrated area under the metabolite peak) change during the cCHRT course, whereas Figure 4 is the box plot representation of the changes during the cCHRT course in the clinical parameters identified by OPLS as significantly affected by the induction treatment.

It is on the basis of these charts that the conclusions of this publication were formulated.

To sum up, the OPLS method is a preliminary analysis, a sieve through which we make a preliminary selection of metabolites. Thanks to this, in the next step we can analyze only important metabolites.

What are the values presented in Fig 2? I understand that they are not equal, but then one can normalize them to appropriate numbers like 1 or percentages. Therefore, it is illogical to present values in a graph that are not presented in a numerical form. Also, what do these triangles mean? Statistical significance? Please write the meaning in the captions.

Answer 2.

We are sorry for our misleading description of Figure 2. Our main plan in this study was to compare the changes that occur in metabolic levels, as well as clinical parameters before and during cCHRT. We initially assumed that since we were analysing the relative concentrations of metabolites, the numerical values on the Y-axis would not be required. However, we agree that this may confuse the readers. Thus, as suggested, we add the numerical values and the units (the relative concentrations are expressed in arbitrary units) in Figure 2.

Yes, the triangles indicate statistical significance. Although this was already explained in the captions of Figures 2 and 4, we also included this information directly in the figures. 

Also please name the y-axis in Fig 3 the same as in Fig 1. At what time the values were taken? Also, is the data presented in Fig 3 also presented in Fig 4? In Fig 4 y axis numbers already appaired. In Fig 4 please write what triangles mean in the graph.

Answer 3.

We apologize for the unclear description of the figure axes and for the incomplete captions under the figures. The captions of Figures 1-4 have been supplemented with the necessary information; In Answer 1 we included full explanations regarding the content of the charts and the changes to their description. Please find this fragment of Answer 1 below:

The OPLS model was used to identify the affected metabolites, we do not differentiate the patients here based on the treatment they received (Fig 1 and 3): both show the coefficients overview plot for the OPLS model, however, Figure 1presents how the X variables (iCHT and duration of cCHRT) affect the blood serum metabolic profile (Y variables), whereas Figure 3 shows how the X variables (duration of cCHRT and iCHT) affect various clinical parameters (Y variables). In turn, the box plots (Fig 2 and 4) compare the changes in the metabolite’s levels occurring over time and the clinical parameters by the treatment received. Figure 2 shows how the relative metabolite concentrations (integrated area under the metabolite peak) change during the cCHRT course, whereas Figure 4 is the box plot representation of the relative changes during the cCHRT course in the clinical parameters identified by OPLS as significantly affected by the induction treatment.

We have also made the appropriate changes to the text of the revised publication.  We added the name of the y-axis in Figure 3. Both OPLS models, the one analysing the metabolites and that analysing the clinical parameters, were built using all collected samples (843 samples collected the day before and during cCHRT). We have added this information in the text of the publication and in the figure captions.

The analysis scheme is identical for the metabolites and the clinical parameters. The OPLS model was used to initially select the variables changing over as revealed in the box plots.

The meaning of the triangles have already been explained in the caption of Figure 4, but to make it distinct we added the legend directly into this figure.

Additionally, we also marked the units for individual clinical parameters.

 In the discussion part, I miss the survival rate of the patients or at least the values of decreased tumours or something like this. Without them, it is just a change in blood value. Of course, it is clear that it will change since the body is intoxicated and affected by ionising radiation. To my understanding there is no novelty there.

The idea behind the study is to compare the toxicities during cCHRT alone and cCHRT with iCHT.

The controversy (or uncertainty) surrounding induction chemotherapy stems from its toxicity that affects more or less all treated patients. Chemotherapy induced adverse events are usually divided into haematological (e.g. leukopenia, neutropenia, anaemia) and non-haematological (e.g. mucositis, nausea, vomiting). At least half of the patients experience severe toxicity during iCHT. Severe toxicity could affect eventual treatment decisions and make subsequent treatment less effective. Therefore, the selection of good candidates for iCHT is a crucial goal.

 According to our review of the literature on the treatment of head and neck cancer, which we have been conducting for several years, the toxicity of iCHT is well documented, but we have not found any publication that directly compares patients with and without iCHT during subsequent stages of treatment in terms of toxicity and tolerability of treatment. This is definitely new and that is why we stressed this point in the introduction of the paper.

The effectiveness of the subsequant  treatment, i.e. radio- or chemoradiotherapy, depends mainly on administering the full planned dose at the planned time. Any interruption of treatment negatively affects the probability of recovery. On the other hand, the probability of cure increases as the intensity of the treatment increases. Until now, patients after iCHT were believed to be at risk and that subsequent treatment should be carefully intensified. The results of our publication indicate that, yes, patients after iCHT start chemoradiotherapy (cCHRT) with a altered metabolic profile and clinical parameters (indicating hematologic toxicities), but this does not affect the tolerability of cCHRT (in the study group, patients undergo cCHRT just like those without iCHT, they do not have interruptions in treatment or other life-threatening events). Moreover, most induction-induced changes disappear after cCHRT. This is again a novelty that has been included in the discussion.

These results clearly indicate that concerns about the use of iCHT itself (due to toxicity and uncertain response rate) and the intensity of the subsequent stages of treatment are not justified.

We agree that evaluating the effectiveness of treatment is extremely important, but this is beyond the scope of this paper. Of course, we conduct such analyses, but these results will be published as a separate article.

Regarding iCHT response, only patients whose tumor response (partial or complete regression), qualifying them for cCHRT treatment, were included in the study group – this information was already included in Section 4.1 (Characteristics of the Patients’ Groups).

In methodology, table 1 is not clear. What is II, III, or IV (probably stages of cancer? Or maybe the age groups?).

Yes, TNM means tumor-node-metastasis international classification system to rank a malignancy. The information in Table 1 has been clarified by adding the term “stage”.

How was the psychological examination done (distress and pain levels)?

 We apologize for the lack of necessary information. The levels of distress and pain were obtained using the Numeric Pain Rating Scale (NPRS) and the Distress Thermometer (DT) – such explanation has been added in 4.2 section.

 In conclusion, the authors state about successive treatment. In which way it was evaluated where is the data about this? The reviewer only can find blood serum and blood cell changes.

We assume that the above comment concerns the opening sentence of the conclusions:

“In summary, this retrospective study shows that the bulky LA-HNSCC patients who received induction chemotherapy begin successive treatment (in this case the concurrent chemoradiotherapy) with significant alterations in the metabolic and clinical profiles”.

The statement "successive treatment" refers to the next stage of treatment after iCHT, which is cCHRT. In this article, we do not discuss the effectiveness or success of overall treatment.

Comments on the Quality of English Language

The English language used in the text is challenging to comprehend. Expressions like 'statistical importance,' 'iCHT was realized with docetaxel-cisplatin and 5-fluorouracil (TPF),' and 'The overnight fasting blood samples from the peripheral vein were collected weekly on each day of the ARS evaluation' make it difficult to follow.

We would like to apologize for using the unclear phrases. The indicated by the Referee expressions have been replaced by the appropriate and clear information.

The term “statistical importance” has been replaced by the “statistical significance”.

Instead of using “'iCHT was realized with docetaxel-cisplatin and 5-fluorouracil (TPF)” we describe the iCHT patients: “The iCHT patients were treated with docetaxel/cisplatin/5-fluorouracil (TPF)”.

The last sentence: “The overnight fasting blood samples from the peripheral vein were collected weekly on each day of the ARS evaluation” has been replaced by “The peripheral venous blood samples were collected once a week (after the patient fasted overnight)”.

Reviewer 2 Report

Comments and Suggestions for Authors

The authors have managed to address all of my queries.

Round 3

Reviewer 1 Report

Comments and Suggestions for Authors

The explanation fitted what I asked. Paper is publishable.

Comments on the Quality of English Language

Quite good